# Dose-Dependent Genetic Resistance to Azole Fungicides Found in the Apple Scab Pathogen

**DOI:** 10.3390/jof9121136

**Published:** 2023-11-24

**Authors:** Thomas Heaven, Andrew D. Armitage, Xiangming Xu, Matthew R. Goddard, Helen M. Cockerton

**Affiliations:** 1National Institute of Agricultural Botany, New Road, East Malling, West Malling, Kent ME19 6BJ, UK; xiangming.xu@niab.com; 2The School of Life and Environmental Sciences, University of Lincoln, Lincoln LN6 7DL, UK; mgoddard@lincoln.ac.uk; 3John Innes Centre, Norwich Research Park, Norwich NR4 7UH, UK; 4Natural Resources Institute, University of Greenwich, Kent ME4 4TB, UK; a.d.armitage@greenwich.ac.uk; 5School of Biosciences, University of Kent, Canterbury CT2 7NZ, UK; h.cockerton@kent.ac.uk

**Keywords:** quantitative trait loci (QTLs), plant pathogen, biotroph, linkage map, genotyping by sequencing, single nucleotide polymorphism (SNP), genome, dose response, polygenic, tebuconazole

## Abstract

The evolution of azole resistance in fungal pathogens presents a major challenge in both crop production and human health. Apple orchards across the world are faced with the emergence of azole fungicide resistance in the apple scab pathogen *Venturia inaequalis*. Target site point mutations observed in this fungus to date cannot fully explain the reduction in sensitivity to azole fungicides. Here, polygenic resistance to tebuconazole was studied across a population of *V. inaequalis*. Genotyping by sequencing allowed Quantitative Trait Loci (QTLs) mapping to identify the genetic components controlling this fungicide resistance. Dose-dependent genetic resistance was identified, with distinct genetic components contributing to fungicide resistance at different exposure levels. A QTL within linkage group seven explained 65% of the variation in the effective dose required to reduce growth by 50% (ED_50_). This locus was also involved in resistance at lower fungicide doses (ED_10_). A second QTL in linkage group one was associated with dose-dependent resistance, explaining 34% of variation at low fungicide doses (ED_10_), but did not contribute to resistance at higher doses (ED_50_ and ED_90_). Within QTL regions, non-synonymous mutations were observed in several ATP-Binding Cassette and Major Facilitator SuperFamily transporter genes. These findings provide insight into the mechanisms of fungicide resistance that have evolved in horticultural pathogens. Identification of resistance gene candidates supports the development of molecular diagnostics to inform management practices.

## 1. Introduction

Apple scab, caused by the fungal plant pathogen *Venturia inaequalis,* is one of the most damaging diseases impacting apple production [1,2,3]. Infections via airborne spores result in characteristic “scab” like lesions on both leaves and fruit; these lesions render fruits unacceptable for direct-to-consumer retail. Extensive control measures are, therefore, required to prevent scab outbreaks and mitigate the severe economic impact of the disease. It is estimated that apple producers in the eastern United States alone spend in excess of USD 18.6 million per year on scab control [4].

Many fungicides are registered for the control of apple scab, and growers are advised to utilise products with differing modes of action to prevent the development of fungicide resistance [5]. Despite this, *V. inaequalis* has evolved resistance to successive classes of fungicides going back to the 1950s [6]. This resistance has often developed quickly after the introduction of a new active and been maintained in the pathogen population thereafter [6]. Combined with tightening regulations on pesticide use, fungicide resistance has left growers reliant on a small number of active ingredients.

Currently, demethylase inhibitors are considered the most effective class of fungicide available for scab control [7]. Sterol DeMethylase Inhibitors (DMIs/azoles) target an essential precursor for ergosterol biosynthesis in fungi (which does not occur in animals): the enzyme 14 α-demethylase, which is encoded by the CYtochrome P450 monooxygenase (CYP) superfamily gene *CYP51* (*ERG11*) [8]. Non-competitive direct binding of DMIs to 14 α-demethylase leads to the accumulation of 14-methylated sterols, which in turn disrupt the cell membrane, slowing fungal growth [9]. DMIs are known for their strong post-infection activity, resulting in a reduction in spore production, and can also prevent scab lesions from developing if applied within 48–72 h of infection [7,10,11,12,13]. Given the limited number of fungicides that are still both available and effective against apple scab, it is necessary to monitor for resistance. If *V. inaequalis* evolves widespread resistance to DMIs before effective substitutes become available, growers stand to suffer substantial economic losses. An understanding of the genetics behind fungicide resistance is an essential prerequisite in order to design effective anti-resistance strategies and ensure the sustainable use of fungicides.

Reducing *V. inaequalis* sensitivity to DMIs has been a concern for apple growers since the mid-1980s, and reports from around the globe indicate that DMI resistance is becoming more prevalent [14,15,16,17,18,19,20,21,22,23,24,25,26,27]. *V. inaequalis* isolates with reduced sensitivity to DMIs were first identified within six years of DMI fungicide deployment in Canada [28]. A build-up of resistance was observed, progressing in a step-like manner and eventually resulting in a complete breakdown of scab control in the tenth year [28]. By contrast, in the eastern United States, it took more than 25 years for DMI-resistant strains of *V. inaequalis* to become established; Cox has subsequently reported that 73 of 93 surveyed orchards in the region could now be considered to have ‘practical resistance’ to the DMI ‘myclobutanil’ [4,29,30]. However, *V. inaequalis* populations considered to be practically resistant to myclobutanil were found to be manageable using other DMIs such as difenoconazole. Resistance to DMI fungicides in scab is widely considered to be incremental, quantitative, and/or dose-dependent [4].

Target-site point mutations of the *CYP51* gene can explain resistance to DMIs in many fungal species, e.g., *Candida albicans* [8,31,32,33,34,35,36,37]. However, *CYP51* point mutations that confer DMI resistance are generally not conserved amongst phytopathogenic fungi [32,38]. Yaegashi et al. were the first to report a point mutation of *CYP51* associated with reduced DMI susceptibility in *V. inaequalis* (Y133F) [39]. However, this mutation could not fully explain variation in DMI sensitivity. Similar observations were made by Hoffmeister et al., who were able to associate several *CYP51* mutations with higher EC_50_ values (half maximal effective concentration) but also found examples of wild-type *CYP51* isolates with increased EC_50_ for DMI fungicides [40]. Thus, it is clear that *CYP51* mutations contribute to DMI resistance in apple scab; however, they cannot fully explain all the resistance observed [34,39,40,41]. Indeed, previous studies have found that, in some orchards, none of the DMI-resistant *V. inaequalis* isolates sampled contained a *CYP51* mutation [41,42]. This suggests that multiple mechanisms of DMI resistance have evolved in *V. inaequalis*.

Cordero-Limon et al. found that the progeny of a cross between fungicide-sensitive and -resistant *V. inaqualis* strains displayed quantitative variation in DMI sensitivity [43]. They concluded that a minimum of two genetic loci were involved in DMI resistance in *V. inaequalis*; however, the basis of this resistance remains uncharacterised. If DMI resistance in *V. inaequalis* is polygenic in nature, this could explain why field isolates exhibit a spectrum of sensitivity levels rather than a binary sensitive and insensitive phenotype [17,41,44]. However, DMI resistance QTLs, outside of the *CYP51* gene or immediate upstream region, have not previously been reported in apple scab [39,40,41,42]. Here, we investigate the hypothesis that *V. inaequalis* resistance to the demethylase inhibitor fungicide ‘tebuconazole’ is a polygenic trait, controlled by genomic regions distinct to those involved in established *CYP51* resistance. To test this, we sequenced the progeny of a cross between resistant and sensitive isolates and mapped genetic loci associated with fungicide resistance.

## 2. Materials and Methods

### 2.1. Generation of a Mapping Population and Measurement of Sensitivity

A *V. inaequalis* mapping population was generated and assayed for fungicide susceptibility by Cordero-Limon et al. [43]. Briefly, a susceptible isolate ‘AF28’ (ED_50_ = 0.12 mg L^−1^ (ED_50_ = the effective dose required to cause a 50% reduction in mycelial growth)), derived from Ash Farm, Worcestershire, UK, and never treated with fungicides, was crossed with a resistant isolate ‘Spartan 1’ (ED_50_ = 3.36 mg L^−1^) derived from a Kent orchard where fungicide resistance had been reported. Crossing of the two isolates was performed by sub-culturing mycelial plugs of each parent onto the same leaf decoction plate, chilled to 4 °C to facilitate sexual reproduction.

A total of 81 progeny strains were assessed for their level of susceptibility to tebuconazole. The commercial fungicide Folicur 25 EW (250 g L^−1^ tebuconazole) from Bayer Crop Science UK, Cambridge, UK, was used to amend PDA media (39 g L^−1^) (Oxoid, Basingstoke, UK), onto which 4 mm diameter circular plugs of each isolate were sub-cultured. Fungal growth was recorded at four fungicide concentrations (4 mg L^−1^, 1 mg L^−1^, 0.1 mg L^−1^, and 0.01 mg L^−1^) as well as on control plates containing no fungicide. These doses captured the declining phase of dose–response curves for both resistant ‘Spartan 1’ and susceptible ‘AF28’ parent isolates and were selected based on discriminatory doses used in previous studies of DMI resistance in *V. inaequalis* [45,46,47,48]. Two perpendicular measurements of colony diameter were taken for each colony three weeks after subculturing. The entire experiment was repeated once; within each replicate, each combination of isolate and concentration was represented by one plate containing three to four mycelial plugs, which were considered pseudoreplicates.

### 2.2. Calculation of Effective Dosages

Effective dose of fungicide required to reduce colony growth by 10, 50, and 90% were estimated for each isolate. The growth area of each colony was estimated, assuming an oval shape and subtracting the initial plug area. From this, mean colony growth measurements were calculated for each treatment within each repeat of the total assessment. These data were transformed into a percentage of maximum growth for each treatment (mean growth on fungicide-free control plates). The upper asymptote of each response curve could then be fixed at 100 (growth unimpeded by fungicide) and lower asymptotes fixed at zero (a lethally high dose). The R package ‘drc’ was used to model the dose–response curve for each isolate [49]. A two-parameter log-logistic model was determined to best-fit the data through assessment of AIC values generated from exploratory fitting of different dose–response models. The ‘ED’ function within the ‘drc’ package was used to estimate absolute ED_10_, ED_50_, and ED_90_ values for each strain (effective dose required for a 10, 50, and 90% reduction in growth, respectively) along with corresponding standard errors and upper and lower limits (Appendix A).

Isolates with fewer than two doses in the declining phase of the response curve were discarded, as it was not possible to model the shape of the curve in this instance. Additionally, one datapoint, for isolate ‘RS11’ at zero fungicide concentration, was discarded as anomalous; this measurement for RS11 was far below the other replicates at zero fungicide concentration and values at higher fungicide doses.

### 2.3. DNA Extraction and Sequencing

DNA extractions were performed via a cetyltrimethylammonium bromide (CTAB)/chloroform–isoamyl alcohol extraction method. Frozen haploid mycelia were first homogenized via grinding in a geno/grinder^®^ (Cole-Parmer, St. Neots, UK) with 2 ball bearings in 2 mL Eppendorf tubes at 1500 rpm in 20 s bursts for 2 min chilling with liquid nitrogen between bursts. Following this, samples were gently mixed at Room Temperature (RT) in 1.5 mL lysis buffer (Appendix A), alone for 30 min, followed by an additional 30 min of incubation after the addition of 20 µL of proteinase K (Qiagen, Venlo, The Netherlands). Samples were then cooled on ice for 5 min, before 250 µL of 5 M potassium acetate was added, cooling then continued for an additional 5 min. Following this, samples were centrifuged for 12 min at 5000× *g*, and the supernatants were transferred to safe lock tubes.

To wash samples, a 1:1 volume of Phenol/Chloroform/Isoamyalcohol 100 mM Tris-EDTA pH 8 (P:C:I) (Thermo Fisher Scientific, Waltham, MA, USA) was added; samples were then inverted for 2 min and centrifuged for a further 10 min at 4000× *g* before transfer of supernatants to fresh tubes. This wash step was repeated three times per sample, followed by a fourth wash using Chloroform/Isoamyl alcohol (C:I) (Sigma-Aldrich, Saint Louis, MO, USA) in place of P:C:I.

To precipitate samples, 200 µL of sodium acetate (3 M, pH 5.2) and 800 µL of isopropanol were added per ~1 mL of supernatant, and samples were inverted at RT for 10 min. Samples were then centrifuged at 8000× *g* for 30 min before supernatants were removed, and the resulting pellets were washed three times in 1.5 mL of 70% ethanol through resuspension, centrifugation at 13,000× *g* for 5 min, and disposal of supernatants. Samples were then left to dry at RT for 30 min before addition of 100 µL of 10 mM Tris pH 8.5 and incubation at RT for 30 min to dissolve the extracted DNA.

Final DNA quality was assessed using a Nanodrop 1000 spectrophotometer (Thermo Fisher Scientific, Waltham, MA, USA). Purification, fragmentation, DNA library construction, and sequencing on an Illumina NovaSeq platform using paired-end chemistry (PE150) were performed by the commercial sequencing company Novogene.

### 2.4. Calling Single Nucleotide Polymorphism Markers

In order to genotype isolates, SNPs were identified using the GATK4 variant calling pipeline [50]. Raw sequencing reads were first trimmed and adapters removed using Trimmomatic v0.39 [51]. Additionally, sequencing depth for each isolate was assessed using the K-mer Analysis Toolkit v2.4.2 function ‘kat plot spectra-cn’ [52]. Trimmed reads were then aligned to the *V. inaequalis* genome generated by Passey et al. (Genbank accession: GCA_003351075.1) using Bowtie 2 v2.4.1 [53,54]. Multimapping reads, discordant reads, and duplicates were removed, and read group and sample names were added to each mapped read using SAMtools v1.1 [53,55]. Following this, Picard tools v2.26.11 was used to create a sequence dictionary for the reference scab sequence, and SNPs were called by GATK ‘HaplotypeCaller’. GATK ‘combineGVCFs’ was used to combine samples and allow joint genotyping via GATK ‘GenotypeGVCFs’.

Vcflib toolkit v0.1.16 and GATK4 were then used to perform filtering and removal of erroneous or low-quality SNPs (successful genotyped in <95% of individuals, quality score of <30 (base call accuracy of 99.9%), read depth of <5, maximum allele frequency of >60%, minimum allele frequency of <40%, or a minor allele count of <3). Following this, call data were filtered using GATK ‘VariantFiltration‘, retaining: QUAL < 30.0; QD < 2.0; SOR > 3.0; FS > 60.0; MQ < 40.0; MQRankSum < −12.5; ReadPosRankSum < −8.0. The resulting SNPs were plotted using R v4.3.0 [56].

### 2.5. Linkage Map Generation

Joinmap 5 was used to calculate a genetic linkage map for *V. inaequalis* [57]. Prior to mapping, SNPs with missing data, or those that were homozygous within the population, were excluded. Additionally, a principal component analysis was performed to identify any rogue genotypes across the isolates. Filtered SNPs were first converted from variant call to HapMap format using Tassel v5.2.81 and, subsequently, from HapMap to raw mapmaker format via a custom Python script (https://github.com/harrisonlab/nano_diagnositcs/blob/master/V_inaequalis/hapmap2mapmaker.py, accessed on 20 November 2023). Mapmaker files were then input to Joinmap 5 [57]. In order to reduce the computational resources required to calculate the linkage map, the ‘exclude similar loci’ function was used to exclude identical SNPs, resulting in a minimal set of 861 informative markers. Ten linkage groups were calculated at LOD = 3.4. The order of SNPs within these groups was then curated to ensure congruity with the known physical position of SNPs in reference genome contigs [54].

### 2.6. Mapping of Quantitative Trait Loci

Each of 861 informative SNPs was tested for association with differences in effective dose between genotypes. To facilitate this, effective dose scores (ED_10_, ED_50_, and ED_90_) were square root transformed to normalise the distribution of phenotypic data before one-way ANalysis Of VAriance (ANOVA) with Benjamini–Hochberg multiple test correction was performed to identify SNP markers significantly associated with effective dose (*p* ≤ 0.05). Significant SNPs were taken forward for QTL identification. QTL regions were defined surrounding a given QTL’s focal SNP (the SNP exhibiting the highest association with the fungicide resistance trait) and delimited by flanking SNPs on either side with comparatively lower associations.

Effect size and Proportion of Variance Explained (PVE) were calculated based upon the focal SNP within QTL regions. To estimate effect size, the difference in mean effective dose between isolates carrying the resistant parent genotype and those carrying the susceptible parent genotype was divided by the standard deviation for isolates carrying the susceptible genotype.
$$\frac{mean_{resistant}-mean_{susceptible}}{standard\;deviation_{susceptible}}$$

PVE was estimated by dividing the sum of squares (SSQ) for effective dose of isolates carrying a variant SNP marker by the sum of squares across all isolates.
$$\frac{SSQ_{variant\;SNP}}{SSQ_{total}}$$

A two-way ANOVA was subsequently performed for the principal markers 009.1_102387 and 005.1_648012 to test for the presence of statistical interactions between QTLs.

### 2.7. Candidate Gene Characterisation

Gene models and functional annotations associated with the reference *V. inaequalis* assembly were used to identify candidate resistance genes within QTL regions [54]. Putative proteins were investigated where genes fell within two QTL regions: firstly, between the SNP markers 009.1_97361 and 009.1_402743 (positions 97,361 and 402,742 of the contig Genbank accession: QFBF01000009.1); and secondly, between SNP markers 005.1_642152 and 005.1_899960 (positions 642,152 and 899,960 of the contig Genbank accession: QFBF01000005.1). Amino acid sequences were aligned to the non-redundant protein sequences (nr) database using the NCBI BLASTp web tool with default parameters to search for homologous proteins [58]. Additionally, protein families, domains, and functional sites were classified by amino acid searches using the European Bioinformatics Institute InterPro and PfamScan web tools with default parameters [59,60].

## 3. Results

### 3.1. Progeny Isolates Segregated by Tebuconazole Sensitivity

Sensitivity to tebuconazole within the crossing population ranged from 0.015 mg L^−1^ to 2.63 mg L^−1^ (measured as ED_50_). The distribution of sensitivity values was continuous across the population, and isolates could not be divided into discrete groups (Figure 1). The fungicide-sensitive parent (AF28) had an ED_50_ of 0.27 mg L^−1^ and ranked as the 24th least sensitive isolate. In contrast, the resistant parent (Spartan 1) had one of the lowest sensitivities within the population (ED_50_ = 1.84 mg L^−1^), second only to one progeny isolate. ‘Spartan 1’ was the least sensitive isolate as measured by ED_90_; however, there were 16 isolates with a higher ED_10_ value. ‘AF28’ had the 18th lowest ED_10_ value and 29th lowest ED_90_ value. ‘Spartan 1’ was consistently less susceptible than ‘AF28’.

### 3.2. Whole-Genome Resequencing Identified Single Nucleotide Polymorphisms across the Mapping Population

Whole-genome skim sequencing was performed for 52 apple scab progeny isolates and the two parents of the mapping population, ‘AF28’ and ‘Spartan 1’. An average of 9,115,913 raw reads were generated for each isolate, with a minimum coverage of ~15× achieved for the 54 isolates, including the two parents. An average of 89.9% of paired trimmed reads aligned to the reference assembly.

A total of 603,595 SNP sites were called across the ~72 Mb *V. inaequalis* reference genome, a SNP, on average, every 119 bp. Of these, a total of 46,727 were high-quality polymorphic SNPs, representing an average of one SNP per 1540 bp interval. These SNPs fell across 90 of the 238 contigs within the reference assembly, with these 90 contigs representing 89% of the total assembly (64,327,219 bp of 72,310,420 bp).

### 3.3. Single Nucleotide Polymorphisms Mapped to Nine Linkage Groups

A total of 861 informative SNP markers were used to generate a linkage map consisting of 10 groups for *V. inaequalis*, which has seven chromosomes. The size of linkage groups ranged from 17,896,836 bp for linkage group one, represented by 241 informative SNPs, to 612,523 bp for linkage group ten, represented by one informative SNP (Table 1).

### 3.4. Two Dose-Dependent Quantitative Trait Loci for Tebuconazole Sensitivity Were Identified

Genomic regions associated with tebuconazole sensitivity were determined through QTL mapping of genotypic and phenotypic data. SNPs significantly associated with reduced tebuconazole sensitivity were identified and mapped to *V. inaequalis* linkage groups, with significance assessed at ED_10_, ED_50_, and ED_90_ levels. From this, two QTLs were identified at ED_10_, each comprising a central ‘focal SNP’ flanked by SNPs whose association diminishes as the genetic distance from the focal SNP increases (Figure 2). Identified focal SNPs fell within linkage group one (LG1) (*p* = 0.044) and linkage group seven (LG7) (*p* = 0.0007).

A dose-dependent genetic response was observed; the QTL in LG7 was found to be significant at both ED_10_ and ED_50_, whilst the QTL in LG1 was only significant at ED_10_. The LG1 QTL had an effect size of −0.9 SD at ED_10_ and explained an estimated 34% of observed variance. The effect size of this LG1 QTL was negative as the resistant parent carried a susceptible variant marker at this QTL. Meanwhile, the LG7 QTL had an effect size of 1.36 SD and explained 36% of the variance in ED_10_; in contrast, 65% of variance in ED_50_ was explained by the LG7 QTL, which had an effect size of 2.18 SD. Given an overall ED_50_ range of 4.71 mg L^−1^, these QTLs were associated with major effect alleles. A third putative QTL in LG4 fell slightly below the threshold for significance (*p* = 0.059) at ED_50,_ with an effect size of 0.64 SD, and explained 26% of the variance.

Interaction between QTLs was investigated. Isolates carrying resistance alleles at both QTLs were observed to have reduced susceptibility at ED_10_ and ED_50_ versus isolates carrying either an LG1 resistance allele or an LG7 resistance allele alone (Figure 3). A two-way ANOVA, considering the focal SNPs for each of the two QTLs, found no evidence of an interaction effect associated with tebuconazole sensitivity (Table 2); this indicates that the two QTLs have an additive effect for resistance at these doses.

### 3.5. Quantitative Trait Loci Location and Gene Content

Genes models were investigated within the two QTL regions predicted. The QTL in LG7 spanned 257 kb of the *V. inaequalis* genome and was associated with a greater difference in fungicide sensitivity than any other locus identified. A total of 70 genes were predicted within this region (Appendix A), including six with InterPro Gene Ontology (GO) term annotations for transmembrane transport. Additionally, three putative proteins returned annotations/sequence similarity matches related to the regulation of transcription. These annotations were investigated in detail as improved efflux of toxins and altered expression of *CYP51* are two possible fungicide resistance mechanisms. Twenty gene models in this QTL region contained non-synonymous SNPs, resulting in amino acid changes in corresponding putative proteins. These included three transmembrane transport genes with the following non-synonymous SNPs: A610T and L615F substitutions in a putative voltage-gated chloride channel protein (Accession: RDI87354.1); a Q1154E substitution in a putative ATP-Binding Cassette (ABC) transporter protein (Accession: RDI87357.1); as well as T489A, H501R and A774E substitutions in a putative active sulfate transmembrane transporter (Accession: RDI87081.1). Additionally, all three transcription-associated genes carried non-synonymous SNPs: P174T and A458T substitutions in a putative zinc finger protein (Accession: RDI87174.1); a H47R substitution in a putative zinc finger protein (Accession: RDI87163.1); and a T403A substitution in a putative helicase (Accession: RDI87242.1).

The LG7 QTL is located 435 kb from the *V. inaequalis CYP51* gene, which encodes 14-α sterol demethylase; the enzyme responsible for catalysing a key demethylation step in the synthesis of ergosterol and the target of demethylase inhibitor fungicides. Cordero-Limon et al. previously sequenced the *CYP51* genes of ‘AF28’ and ‘Spartan 1’, confirming that there were no point mutations within the gene sequences of the parent isolates [13]. The most significant marker in the LG7 QTL, ‘005.1_648012’, is located a large distance (683 kb) upstream of the *CYP51* gene location. Although SNPs between marker ‘005.1_648012’ and *CYP51* were significantly associated with tebuconazole resistance, this was to a lesser extent than the focal SNP and in a pattern consistent with linkage to a neighbouring allele, indicating the LG7 QTL represents a novel resistance locus.

The second resistance QTL was identified in LG1 and spanned 305 kb; 39 gene models were predicted within this QTL region, 19 of which carried non-synonymous SNPs (Appendix A). Three of these genes encoded putative Major Facilitator Superfamily (MFS) transporter proteins, two of which carried SNPs resulting in amino acid changes: an I796M substitution in the protein Vi05172_g3976 (Accession: RDI85792.1) as well as V223G and G326D substitutions in Vi05172_g3984 (Accession: RDI85945.1). No transcription-related annotations were identified within the LG1 QTL.

Taken together, we identify 39 genes that demonstrate non-synonymous mutations within QTL regions. Through the identification of putative transmembrane transport proteins and transcription factors, we prioritise a subset of these as candidates that may contribute to resistance for future study. These include six gene candidates for LG7 QTL resistance and two gene candidates for LG1 QTL resistance.

## 4. Discussion

Here, we have performed the first QTL mapping study of fungicide resistance in *V. inaequalis* and demonstrated that multiple loci are associated with resistance to the DMI tebuconazole. These results support our original hypothesis that DMI resistance in the apple scab pathogen population is a polygenic trait [43]. Moreover, both genetic loci identified were observed to meditate dose-dependent resistance. Finally, we identify putative transporter proteins in both QTL regions, which may underly this novel dose-dependent resistance mechanism in *V. inaequalis*.

### 4.1. Multiple Loci Contribute to the Level of Tebuconazole Sensitivity in the Apple Scab Pathogen

A progressive increase in resistance to DMI fungicides has been reported in apple scab pathogen populations since the mid-1980s [14,15,17,61]. The QTL mapping results presented here support the hypothesis that, at lower dosages, DMI resistance in *V. inaequalis* is due to the additive effects of multiple loci (Figure 3, Table 2). This may explain the relatively delayed development of DMI resistance in *V. inaequalis* compared to resistance against other chemical families [6]. The evolution of polygenic resistance typically takes longer to arise than monogenic resistance, as at each step, any fitness disadvantage conferred by a resistance mutation must be compensated for, and each iterative step must be advantageous for the mutation to persist [62].

Those QTLs identified here in *V. inaequalis* are dose-dependent; a resistance QTL was identified in linkage group one, which was important only at lower fungicide concentrations (ED_10_), whilst a QTL within linkage group seven was associated with resistance at both ED_10_ and ED_50_. Whereas there were 23 progeny (42.6%) with greater susceptibility than the susceptible parent ‘AF28’ (ED_50_), only one isolate was more resistant than the resistant parent ‘Spartan 1’ (ED_50_). This asymmetric distribution is inconsistent with purely additive QTL effects [43]. Additionally, dominance effects between alleles are unlikely, given that *V. inaequalis* is haploid outside of a very brief diploid stage during sexual reproduction. Epistatic effects are, therefore, likely to underpin the distribution of susceptibilities at higher fungicide concentrations. This asymmetric distribution was present only in ED_50_ and ED_90_ estimates and was not observed for ED_10_ (Figure 1). Therefore, the data and analyses show that resistances associated with the LG1 and LG7 QTLs behave in an additive fashion at low fungicide concentrations, whereas at high fungicide concentrations, only the LG7 QTL, carried by ‘Spartan 1’, has an impact. In a single-gene scenario, no progeny isolates are expected to significantly exceed the parental ED_50_.

The resistant parent ‘Spartan 1’ carries a susceptible variant genotype for the LG1 QTL, whilst the susceptible parent ‘AF28’ has a resistant genotype at this locus. This explains why, for ED_10_, certain progeny isolates were observed to exhibit higher levels of resistance than the resistant parent, whilst others displayed greater susceptibility than the susceptible parent. This result underscores the complexity of DMI fungicide resistance in the apple scab pathogen. Additionally, several loci were identified that appeared to be associated with resistance but fell short of statistical significance in this study, for example, within linkage group four at ED_50_ and linkage group three at ED_10_ (Figure 2). Clearly, there is scope for evolution of the pathogen to further increase the resistance levels of even the most tolerant isolates. This study demonstrates polygenic resistance in a mapping population generated from the isolate ‘Spartan 1’, which was sampled from a single UK orchard suffering from fungicide-resistant scab outbreaks. As such, further research is required to assess the prevalence of the observed resistance across field populations in the UK and globally. Furthermore, it will be important to determine whether the resistance and SNPs described here co-occur with SNPs identified in previous studies (e.g., *CYP51* Y133F) [39,40]. Such monitoring and appropriate mitigation measures are particularly important for sexual species, such as *V. inaequalis*, to prevent the stacking of resistance mutations in pathogen populations.

### 4.2. Quantitative Trait Loci Indicate a Novel Mechanism of Resistance

Azole resistance in fungal pathogens has been attributed to several mechanisms, including point mutations in *CYP51*, duplication of *CYP51*, overexpression of *CYP51*, modified efflux through ABC transporters, alternative modes of sterol biosynthesis, and changes in membrane composition [8,31,32,33,34,35,36,37,38,39,40,41,63,64,65,66,67,68,69,70]. Of these, point mutations and altered expression of the *CYP51* gene and upstream region have previously been investigated in *V. inaequalis* [39,40,41,42]. Mutations in *CYP51* can explain DMI resistance in many fungal species, including plant and human pathogens such as *Aspergillus fumigatus*, *Blumeria graminis*, *Candida albicans*, *Cercospora beticola*, and *Erysiphe necator* [32,34,39,40,41]. However, none of the common *CYP51* point mutations associated with DMI resistance were identified in the resistant parent ‘Spartan 1’ (e.g., Y133F). Of the two resistance QTLs identified in this study, one is in a separate linkage group to *CYP51*, whilst another, strongly associated with resistance at both ED_10_ and ED_50_, is located 435 kb from the *CYP51* gene. This lack of proximity to the *CYP51* gene indicates that, in these QTLs, we have identified discrete alleles associated with resistance.

Insertions < 1000 bp upstream of *CYP51* have previously been associated with resistance to the DMI difenoconazole in *V. inaequalis* [41,42]. However, regulatory elements were not anticipated at a distance of 435 kb or more from the *CYP51* gene, as all DMI resistance mutations reported in *V. inaequalis* to date have been within the gene coding region itself or immediately upstream (<2 kb). Nonetheless, we cannot rule out the possibility that the QTLs identified here represent distal regulatory elements of *CYP51.* Indeed, putative proteins encoded in one QTL returned GO terms related to DNA-templated transcription and had sequence similarity to transcription factor Zn [43]. Additionally, many genes in the QTL regions are of unknown function, and several lacked homologs outside of *V. inaequalis.* These genes may encode a trans-regulatory transcription factor that affects *CYP51*, or they play a role in regulating other genes, leading to resistance.

Overexpression of *CYP51* has been implicated in DMI fungicide resistance in several phytopathogenic fungi [31,71,72,73,74,75,76]. In *Erysiphe necator*, increased *CYP51* copy number enhances fungicide resistance, whilst in *Aspergillus fumigatus*, a second gene encoding the CYP51 protein has been discovered [69,70]. Increased expression of the *CYP51* gene has also been investigated in *V. inaequalis*; Villani et al. (2016) identified a 169 bp repeatable element carrying transcription factor binding sites upstream of *CYP51,* which was associated with a 9–13-fold increase in gene expression and resistance to the DMI difenoconazole [42]. However, neither the presence of these insertions nor *CYP51* overexpression could be associated with myclobutanil resistance, suggesting that different mechanisms may influence resistance to different DMI-class fungicides [42].

### 4.3. Dose-Dependent Resistance May Result from Variant Efflux Transporter Genes

A resistance QTL was identified in LG1, which was important only at lower fungicide concentrations (ED_10_). Within this QTL, two genes encoding putative MFS transporter proteins contained non-synonymous SNPs, resulting in changes to the translated protein sequence and potentially protein function. Upregulation, altered specificity, or altered efficiency of efflux proteins, that detoxify the fungus at lower quantities of fungicide but are overwhelmed at higher concentrations, could explain the dose-dependent nature of resistance. Alterations to transport efflux could also explain the cross-resistance between different DMI fungicides, as previously observed between myclobutanil and tebuconazole in the ‘AF28’ × ‘Spartan 1’ population by Cordero-Limon et al., enabling detoxification of several different fungicides [44,72]. Active efflux transporters, such as ABC and MFS proteins, have remarkably broad substrate specificity, and upregulation of ABC and MFS transporters is known to be involved in the development of resistance to DMIs in other fungi [63,64,77,78,79,80]. For example, enhanced efflux adaptations have been observed in fungicide-resistant strains of *Candida* spp., *Botrytis cinerea*, and *Z. tritici* [81,82,83,84,85,86]. Additionally, laboratory mutants of *V. inaequalis* have been generated which exhibit increased efflux efficiency associated with DMI resistance [87]. Vijaya Palani and Lalithakumari were able to generate DMI-resistant *V. inaequalis* isolates through chemical mutagenesis, resulting in an increased energy-dependent efflux rate [87]. However, changes in chemical transport have not yet been associated with *V. inaequalis* fungicide resistance in the field. As such, transporter genes carrying non-synonymous SNPs in this study represent good resistance gene candidates for future investigation. We cannot rule out the contribution of the other genes possessing non-synonymous SNPs within this region. However, we suggest prioritisation of the identified putative transporter genes for functional validation to assess their role in resistance.

The QTL within LG7, which is associated with resistance at both ED_10_ and ED_50,_ also contained genes encoding putative transporters. However, in this QTL, only one putative transporter protein had broad specificity: an ABC transporter that carries amino acid substitutions in the resistant parent ‘Spartan 1’. This protein, Vi05172_g2818, has sequence similarity to ABC multidrug transporter-like proteins present in many other species and may underly the resistance associated with the LG7 QTL. Whist there is little literature concerning the function of ABC transporters in *V. inaequalis,* ABC transporters have been demonstrated to be involved in resistance to azoles in other species, such as *Penicillium digitatum* and *Botrytis cinerea* [64,78,88].

Unfortunately, if resistance indeed derives from improved efflux transporter activity, this may reduce the effectiveness of mixed fungicide applications combining different modes of action. Cross-resistance is most common between chemicals with the same target site (i.e., Other DMIs); however, enhanced efflux can also cause cross-resistance between different fungicide classes [43,89,90].

### 4.4. Alternative Resistance Mechanisms

In addition to transporter proteins and transcription factors, a number of additional protein families may play a role in DMI resistance. It has been proposed that in addition to CYP51, other proteins in the ergosterol biosynthesis pathway could be involved in resistance to demethylase inhibitors. A reduction in the total rates of lipid and ergosterol synthesis may allow efflux at normal rates to prevent the build-up of damaging levels of methylated ergosterol. In *Ustilago maydis*, a defective desaturase protein in the sterol pathway has been associated with resistance to demethylase inhibitors [66]. In *Z. tritici*, another methyltransferase in the ergosterol biosynthesis pathway (ERG6), a phospholipid biosynthesis protein, and a polyketide synthase involved in melanisation (PSK1) are proposed to be involved in resistance; it is believed that melanin may be able to adhere to fungicidal chemicals, preventing them from reaching their target [63]. In *C. beticola*, a group of polyketide synthase genes have been associated with resistance to the DMI tetraconazole and are believed to cause resistance by fortifying cell membranes [67]. As such, similar mechanisms should also be considered for genes in the vicinity of the LG1 and LG7 QTLs. Such annotations were not observed; however, given that many genes lack any functional annotations (Appendix A, 14 of 39 genes containing non-synonymous SNPs), as gene annotation pipelines improve, we may prioritise additional gene candidates for functional assessment of contribution to DMI resistance.

### 4.5. Lab-Based vs. In-Field Assessments

In this work, fungicide susceptibility was assessed under lab conditions, care must therefore be taken when relating the observed resistance to the field. We note that the recommended tebuconazole dose for in-field apple scab control is 125 mg L^−1^, with the application of 1500 L ha^−1^. However, fungicide exposure under spray conditions differs from exposure under laboratory conditions in infused agar plates. We found that a maximum experimental dose of 4 mg L^−1^ was sufficient to reduce the growth of ‘Spartan 1’, which was sampled from a resistant field population, by ~60% in a laboratory setting. This level of fungicide exposure is in line with similar lab-based studies. For example, using a discriminatory tebuconazole dose of 3 mg L^−1^ in vitro, Keinath et al. were able to predict cross-resistance of *Stagonosporopsis citrulli* isolates to five DMI fungicides *in planta*, whilst Chen et al. used a discriminatory dose of 5 mg L^−1^ to investigate tebuconazole resistance in 1118 *Fusarium graminearum* strains [91,92].

### 4.6. Implications for Practical Management

The identification of resistance gene candidates underpins the development of molecular diagnostics to inform management practices. Ideally, steps should be taken to tackle the emergence of fungicide resistance whilst the frequency of resistance genes islow, as it may be possible to prevent the fixation of resistance alleles in pathogen populations [89,93]. To achieve this, pre-emptive screening for the emergence of resistance alleles is required. Historically, identification of apple scab has primarily relied upon visual inspection of symptoms or spores; however, more recently, molecular techniques such as real-time PCR and loop-mediated isothermal amplification have been applied to *V. inaequalis* detection [94,95,96,97,98]. These methods could be adapted to screen for fungicide resistance markers and monitor the build-up of incremental gains in fungicide resistance within pathogen populations. Low-dose fungicide resistance (e.g., LG1 QTL) may be hard to track in the environment through phenotyping alone. Identifying genetic markers underpinning this resistance is, therefore, an important step towards effective resistance monitoring.

*V. inaequalis’* ability to rapidly evolve and overcome host resistance genes and fungicides is aided by its two reproductive phases. Repeated cycles of asexual reproduction in each growing season generate an enormous number of progeny and facilitate the rapid spread of adaptive mutational events, whilst sexual reproduction over winter facilitates the combination of novel mutations and can accelerate evolution [99]. As growers seek to maximise the effective lifespan of available fungicides and forestall the evolution of resistance, a key principle is to ‘hit hard and hit early’ [100]. This approach minimizes population sizes, thereby limiting opportunities for adaptive mutation, and prevents a gradual buildup of resistance driven by the selective pressure of sublethal fungicide doses [89]. Past fungicide application strategies for the control of apple scab have come under scrutiny where regimes have not adhered to this approach [93]. Strategies that rely upon the strong post-infection activity of DMIs and seek to reduce the quantity of fungicide used, by spraying ‘curatively’ once the pathogen population is well established, on alternate rows of trees, and/or at reduced doses, are to be discouraged [7,10,11,12,13,93]. The identification here of dose-dependent resistance mechanisms in *V. inaequalis* underscores the importance of applying the full recommended dose [101]. In theory, the use of high doses should prevent selection for quantitative resistance by killing partially resistant genotypes along with sensitive isolates, although there is limited evidence of this in field settings [93,101,102].

### 4.7. Summary

We have demonstrated, for the first time, that DMI resistance in *V. inaequalis* is controlled by multiple loci. We also show that a dose-dependent genetic resistance response exists against azole fungicides, which provides insight into the evolution of fungicide resistance in *V. inaequalis* and other fungi that are subject to azole fungicides. These findings represent an important step towards understanding the complexity of resistance and improving the control of the apple scab pathogen [96,103]. The identification of novel resistance QTLs highlights putative fungicide resistance genes. Functionally validated resistance genes represent good diagnostic targets for monitoring and surveillance of fungicide resistance in pathogenic fungi, which aids in the early detection and containment of resistant strains [104,105]. Additionally, deciphering the mechanisms behind resistance can be used to model the rate and pattern of resistance development over time. By modulating dose, or choosing combinations of fungicides with different targets or modes of action, it may be possible to overcome or delay the establishment of resistance [102,106].

## Figures and Tables

**Figure 1 jof-09-01136-f001:**
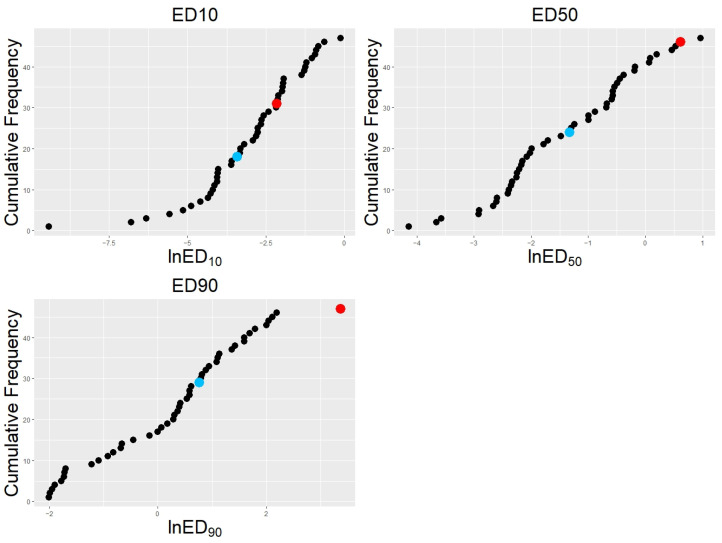
Cumulative frequency of *Venturia inaequalis* tebuconazole effective doses: Sensitivity of 47 apple scab isolates plotted with reducing susceptibility to tebuconazole from left to right; susceptibility defined as the effective dose required to cause a 10, 50, or 90% reduction in mycelial growth: ED_10_, ED_50_, and ED_90_. Progeny isolates are shown in black; parent isolates are shown in blue (Sensitive parent ‘AF28’ with lower effective dosages) and red (resistant parent ‘Spartan 1’ with higher effective dosages). Raw data were generated as part of previous work by Cordero-Limon et al. [43].

**Figure 2 jof-09-01136-f002:**
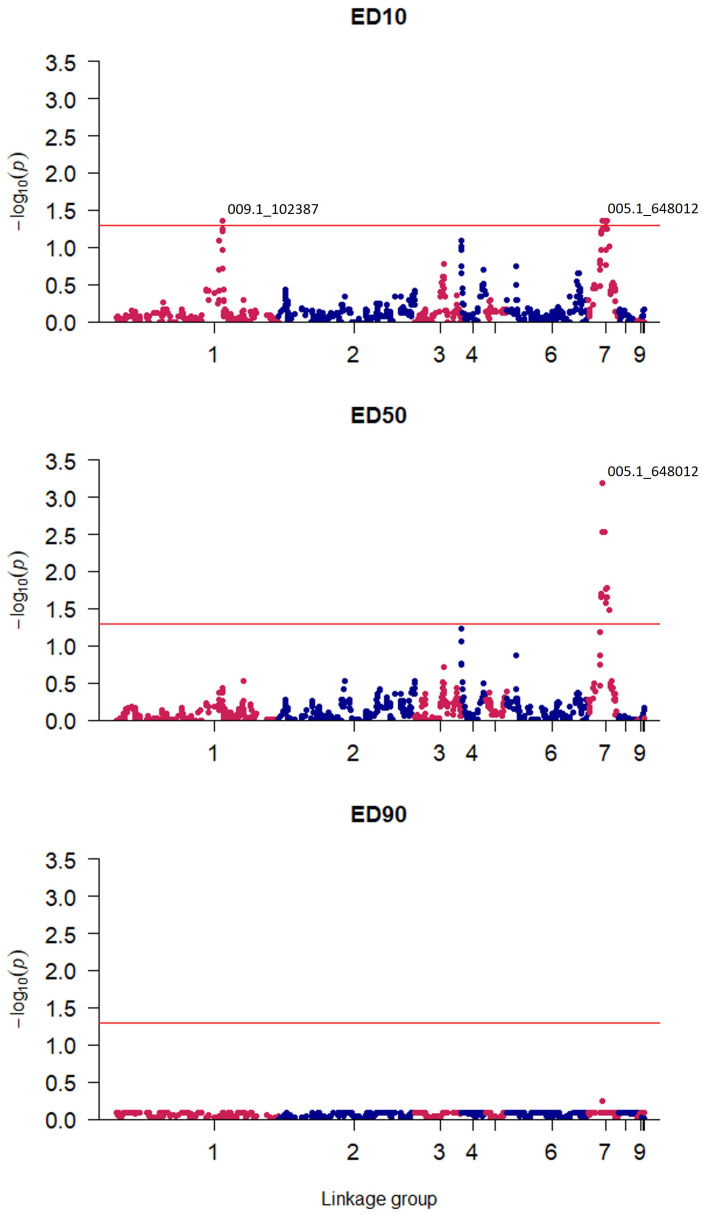
Quantitative Trait Locus (QTL) mapping identifies a dose-dependent response to tebuconazole sensitivity over ED_10_, ED_50_, and ED_90_: 861 informative Single Nucleotide Polymorphism (SNP) markers are plotted by significance of their association at a given effective dose (ED_10_, ED_50_, or ED_90_) against their estimated position within the *Venturia inaequalis* genome in one of the ten linkage groups. Significance of *p* = 0.05 is indicated by a red horizontal line. The ID of the most significant SNP marker is indicated for each identified QTL.

**Figure 3 jof-09-01136-f003:**
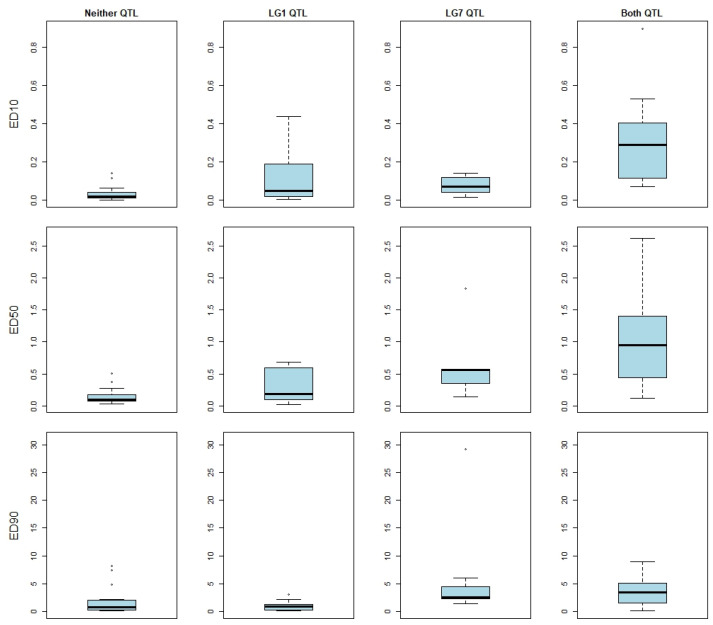
Resistance Quantitative Trait Loci (QTLs) contribute additively to tebuconazole susceptibility at low doses: Contribution of combinations of QTLs to tebuconazole susceptibility, defined as the effective dose required to cause a 10, 50, or 90% reduction in mycelial growth (ED_10_, ED_50_, and ED_90_). Sensitivities for apple scab isolates carrying neither resistant allele (n = 7), carrying only a linkage group one resistance allele (n = 12), carrying only a linkage group seven allele (n = 16), and carrying both resistant alleles (n = 12) are plotted left to right, respectively.

**Table 1 jof-09-01136-t001:** *Venturia inaequalis* linkage groups: The sizes of 10 assembled linkage groups are given along with the number of Single Nucleotide Polymorphisms (SNPs) these contain and the number of informative SNPs representing segregation.

Linkage Group	Size (bp)	SNPs	Informative SNPs
1	17,896,836	12,003	241
2	15,525,007	10,572	201
3	6,231,899	5506	84
4	3,556,600	2427	53
5	2,751,175	2122	33
6	9,505,949	7500	147
7	4,036,205	3407	51
8	2,557,858	2250	33
9a	1,643,167	588	11
9b		322	6
10	612,523	30	1

**Table 2 jof-09-01136-t002:** Assessment of non-additive interactions between Quantitative Trait Loci (QTLs): Two-way ANalysis Of VAriance (ANOVA) for linkage group one (LG1) and linkage group seven (LG7) QTLs. *p*-values for LG1 QTL, LG7 QTL, and interaction (LG1 QTL: LG7 QTL) are given for effective dose required to cause a 10, 50, or 90% reduction in mycelial growth (ED_10_, ED_50_, and ED_90_), as well as the percentage of variability explained by these two QTLs, determined using the R-squared value.

Dose	LG1 QTL	LG7 QTL	LG1 QTL:LG7 QTL	Variability Explained (%)
**ED10**	5.56 × 10^−5^	5.41 × 10^−4^	0.19	45.42
**ED50**	5.09 × 10^−3^	9.09 × 10^−6^	0.69	44.31
**ED90**	0.54	1.18 × 10^−3^	0.66	22.73

## Data Availability

The data presented in this study are openly available in NCBI SRA archive, BioProject accession PRJNA817384 Venturia inaequalis Map.

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
