# Peer review of "Dose-Dependent Genetic Resistance to Azole Fungicides Found in the Apple Scab Pathogen"

_jof, 2023, doi:10.3390/jof9121136_

Round 1

Reviewer 1 Report

Comments and Suggestions for Authors

The authors present an interesting premise and they address it well.

I have still some suggestions/questions:

-I suggest the authors to finish the introduction in a more conclusive way, indicating the majors finding of the study, instead of just a description of the work performed

-The writing the materials and methods section should be reviewed to make it more fluid and easier to read

-I advise the authors to exploit more the function of the genes in the identified QTL regions and to further discuss it.

Specific suggestions:

-Line 215: Where it reads “The sum of squares for the SNP marker was then divided by the total sum of squares, giving the proportion of the total variation in effective dose that could be explained by a given SNP marker.”, the authors could show an equation instead.

-Figure 1: The authors should consider depicting AF28 and Spartan 1 in different colours to easily distinguish them

-Lines 293-303: While the affirmations between these lines make sense and are meaningful for this study, they should be in the discussion and not in the results section

-Figure 2: I advise the authors to not keep this figure or to reference it in the results section and explain it better in text.

Comments on the Quality of English Language

The writing of the manuscript should be reviewed to eliminate some punctual English inaccuracies and to make the text more fluid, especially in the materials and methods section

Author Response

We would like to thank the reviewers for taking their time to review this manuscript and for the considered feedback provided. We have addressed and provided a response on all points raised, as detailed below. 

Reviewer 1:

Comments and Suggestions for Authors:

-I suggest the authors to finish the introduction in a more conclusive way, indicating the majors finding of the study, instead of just a description of the work performed

We appreciate the review’s suggestion here. However, we believe that this represents a stylistic preference. As it is, the manuscript introduction ends by outlining the limits of current knowledge and clearly outlining the hypothesis for the study, which directly leads on to the materials and methods section.

It is our understanding that it is not common practice to reiterate major findings of the study in the introduction section for Journal of Fungi articles. Please note that the major findings of this study are outlined for ease of access in the abstract of the manuscript.

-The writing the materials and methods section should be reviewed to make it more fluid and easier to read

We acknowledge the reviewers concerns and have edited the methods section throughout to improve clarity. In particular, details regarding preparation of lysis buffer in section 2.3 have been moved to a supplementary table to improve fluidity.

-I advise the authors to exploit more the function of the genes in the identified QTL regions and to further discuss it.

We thank the reviewer for this comment. Within text, we have highlighted those genes within QTL regions that have sequence similarity to transporter or transcription-related genes. We cannot find evidence of existing publications that explore the functions of the specific genes identified in V. inaequalis. We have edited the end of section 4.3 to convey this (lines 492-495). Additional lab-based functional characterisation of genes is beyond the scope of this study but is highlighted as a priority for subsequent work.

Specific suggestions:

-Line 215: Where it reads “The sum of squares for the SNP marker was then divided by the total sum of squares, giving the proportion of the total variation in effective dose that could be explained by a given SNP marker.”, the authors could show an equation instead.

Equations have been added to improve the clarity of section 2.6 (lines 226-234).

-Figure 1: The authors should consider depicting AF28 and Spartan 1 in different colours to easily distinguish them

The susceptible parent AF28 has been changed to a different colour to allow easy differentiation from the resistant parent Spartan1 in red (Figure 1) (lines 275-282).

-Lines 293-303: While the affirmations between these lines make sense and are meaningful for this study, they should be in the discussion and not in the results section

These two paragraphs have been moved to discussion section 4.1 (lines 399-408).

-Figure 2: I advise the authors to not keep this figure or to reference it in the results section and explain it better in text.

We have taken the reviewers suggestion to improve clarity as we believe that inclusion of Manhattan plots such as this are standard across GWAS/QTL mapping studies.

This figure represents the main findings of the paper, illustrating the dose dependant genetic control of fungicide resistance at ED10, ED50 and ED90. Accordingly, results section 3.4 has been edited (lines 292-296), along with the Figure 2 title (lines 308-309), to clarify the use of this figure for identification of focal SNPs and to allow readers to independently draw conclusion that these represent True QTL.

Comments on the Quality of English Language:

-The writing of the manuscript should be reviewed to eliminate some punctual English inaccuracies and to make the text more fluid, especially in the materials and methods section

The materials and methods section has been edited to improve clarity, preparation of lysis buffer in section 2.3 has been moved to a supplementary figure to improve fluidity. Authors will be happy to further address any additional specific grammatical errors.

Reviewer 2 Report

Comments and Suggestions for Authors

Review Heaven et al. 2023

The manuscript from Heaven et al. describes a Quantitative trait locus which affects sensitivity towards azole fungicides, specifically DMIs targeting ERG11 in lines of the apple pathogen Venturia inaequalis.

Mutations in the gene for ERG11 contribute to much, albeit not all observed DMI resistance.

Using 52 strains from a mapping population of Venturia strains with quantitative variations in DMI resistance and genome DNA sequencing the authors identified two QTLs associated with this trait.

The data are very well presented, the reading was easy and as far as I can judge, the DNA sequencing, mapping and SNP calling was state of the art.

Technically the works is very good, indeed.

The authors provide some speculative hypothesis which of the observed 19 SNPS in the 39 genes at QTL 1 may be relevant, concentrating on MFS transporters. Also for the QTL on linkage group 7 which spans 257 kb and contains 70 genes they found SNPs in possible candidates genes encoding transporters but also transcription factors. 

None of the potential canidates was further analysed, which would be beyond the scope of the paper.

Specific comments:

L352: pleas clarify whether or not the following words relate to unpublished work: “In follow-up to these results, we have identified putative transporter proteins in both QTL regions which may underly 353 a novel dose-dependent resistance mechanism in V. inaequalis.“

Nice work that could be published as is.

Author Response

We would like to thank the reviewers for taking their time to review this manuscript and for the considered feedback provided. We have addressed and provided a response on all points raised, as detailed below.

Reviewer 2:

Comments and Suggestions for Authors:

Specific comments:

-L352: please clarify whether or not the following words relate to unpublished work: “In follow-up to these results, we have identified putative transporter proteins in both QTL regions which may underly a novel dose-dependent resistance mechanism in V. inaequalis.“

We thank the reviewer for highlighting this ambiguous wording. This section does not relate to unpublished work, this sentence has been edited accordingly, section 4 (lines 380-382).

Reviewer 3 Report

Comments and Suggestions for Authors

The study is informative, providing significant insights into the phenomenon of dose-dependent genetic resistance in V. inaequalis. However, a more detailed account of the methodologies employed, particularly the genotyping by sequencing, would enhance the clarity and comprehensiveness of the study, enabling other researchers to replicate the findings more effectively.

The introduction appears overly extensive and could benefit from condensation. Please consider succinctly presenting the core concepts and background information, relegating any supplementary details to the discussion section.

 The hypothesis needs clearer articulation. It’s currently ambiguous regarding the study’s intentions and objectives, and a more precise definition of what was aimed to be investigated or proven would be beneficial.

The representation of data and the chosen statistical models should be examined more rigorously to ascertain the reliability and validity of the findings.

Include a separate paragraph detailing the statistical analysis methods employed in this study.

More explicit justification and elaboration regarding the chosen exposure levels of tebuconazole should be given or discussed. Clarification on how these levels were determined and whether they were varied systematically should be added

The paper should delve deeper into the potential impact and relevance of non-synonymous mutations on fungicide resistance.

The implications and applications of the identified genetic components and mutations on practical management strategies in apple orchards need to be explored more thoroughly.

It is essential to ensure that the study’s findings are appropriately contextualized within the current understanding of azole resistance in fungal pathogens.

The conclusions drawn in the paper should be more closely aligned with the presented data. A comprehensive discussion on the limitations of the study and potential directions for future research should be added

Elucidate more on the observed mutations in ATP-Binding Cassette and Major Facilitator SuperFamily transporter genes, providing a clearer link between these mutations and their role in fungicide resistance.

The identification of resistance gene candidates is a crucial step forward; however, more emphasis on how these findings can support the development of molecular diagnostics to inform management practices is warranted.

Figure S1 and Table S1 should be in main text not as supplementary

Comments on the Quality of English Language

Minor editing of English language required

Author Response

We would like to thank the reviewers for taking their time to review this manuscript and for the considered feedback provided. We have addressed and provided a response on all points raised, as detailed below.

Reviewer 3:

Comments and Suggestions for Authors:

-The study is informative, providing significant insights into the phenomenon of dose-dependent genetic resistance in V. inaequalis. However, a more detailed account of the methodologies employed, particularly the genotyping by sequencing, would enhance the clarity and comprehensiveness of the study, enabling other researchers to replicate the findings more effectively.

We thank the reviewer for their positive comments. We note the lack of clarity in the Materials and Methods section. We have addressed this, in line with concerns also raised by Reviewer 1. The materials and methods section has been edited to improve clarity and better convey the methodology used, section 2 (throughout).

-The introduction appears overly extensive and could benefit from condensation. Please consider succinctly presenting the core concepts and background information, relegating any supplementary details to the discussion section.

The introduction section has been shortened, including moving some text to the discussion section (lines 550-554). We believe that it is important to provide the current brief-introductions to V. inaequalis, the chemical action of azole fungicides, and the development of resistance to azoles in V. inaequalis, rather than assuming prior knowledge in any of these areas. This allows access by a readership across applied horticulture, fungal-pathology and those studying fungicide resistance more broadly.  The length of the revised introduction is now in-line with other recent publications in Fungal Biology.

 -The hypothesis needs clearer articulation. It’s currently ambiguous regarding the study’s intentions and objectives, and a more precise definition of what was aimed to be investigated or proven would be beneficial.

We appreciate the opportunity to clarify core objectives. We have provided a clear statement of this study’s hypothesis in section 1 (lines 97-99), 4 (lines 377-378), 4.1 (lines 385-387).

-The representation of data and the chosen statistical models should be examined more rigorously to ascertain the reliability and validity of the findings.

We have added text to address this feedback. Details have been added to sections 2.2 (lines 128-139) and 2.6 (lines 206-229) which cover the statistical models used in the study. Discussion of the studies limitations has been added in sections 4.1 (lines 418-424), 4.3 (lines 485-487) and  4.5 (lines 522-534).

-Include a separate paragraph detailing the statistical analysis methods employed in this study.

In response to this feedback, we have provided additional detail within the manuscript. Statistical methods used in this study are detailed sections 2.2 (lines 128-139) and 2.6 (lines 206-229) describing, respectively, how effective dose phenotypes were modelled from raw measurements of fungal growth, and how the relationship between genotype and phenotype was investigated. We also highlight that the materials and methods section has been edited throughout to improve clarity in response to other reviewer comments, including the addition of equations to section 2.6.

-More explicit justification and elaboration regarding the chosen exposure levels of tebuconazole should be given or discussed. Clarification on how these levels were determined and whether they were varied systematically should be added

We agree that detail was lacking in the original draft. We have consequently clarified the selection of exposure levels of tebuconazole. The Materials and methods section 2.1 (lines 117-120) has been edited to clarify the reasoning behind the selected exposure levels, we additionally discuss the relationship of those doses used to field rates of fungicide in section 4.5 (lines 522-534).

-The paper should delve deeper into the potential impact and relevance of non-synonymous mutations on fungicide resistance.

In response to the reviewer’s comments, we have clarified the manuscript text. Identification of non-synonymous mutations to efflux transporters and transcription factors formed the basis of our identification of candidate resistance genes in this study. This has been clarified in section 3.5 (lines 337-342, 348, 369-373) and section 4.3 (lines 466-471, 474-479, 484-494, 487-492). The GO term annotations for all genes in the QTLs are also given in Table S2.

-The implications and applications of the identified genetic components and mutations on practical management strategies in apple orchards need to be explored more thoroughly.

We have now added Section 4.5 (lines 535-568) to further discuss the implications upon future diagnostic and management practices.

-It is essential to ensure that the study’s findings are appropriately contextualized within the current understanding of azole resistance in fungal pathogens.

Section 4.2 (lines 428-437, 453-463) has been edited to clarify current understanding of azole resistance in fungal pathogens generally vs. what is known/has been found in V. inaequalis.

-The conclusions drawn in the paper should be more closely aligned with the presented data. A comprehensive discussion on the limitations of the study and potential directions for future research should be added

Discussion of the studies limitations and potential future research has been added in sections 4.1 (lines 418-426), 4.3 (lines 485-487) and 4.5 (lines 522-534).

-Elucidate more on the observed mutations in ATP-Binding Cassette and Major Facilitator SuperFamily transporter genes, providing a clearer link between these mutations and their role in fungicide resistance.

The relevance of the identified non-synonymous SNPs to potential resistance mechanisms has been clarified in section 4.3 (lines 466-470, 474-479). As referenced in the manuscript ABC and MFS transporters are known to be involved in DMI resistance in a range of other species such as Candida spp., P. digitatum, Z. tritici and B. cinerea (lines 477-479, 494-497).

-The identification of resistance gene candidates is a crucial step forward; however, more emphasis on how these findings can support the development of molecular diagnostics to inform management practices is warranted.

We agree that there is scope for molecular diagnostics to track fungicide resistance within pathogen populations. Section 4.5 (lines 535-568) has been added to emphasise further the application of these findings for the development of diagnostics and management practices.

-Figure S1 and Table S1 should be in main text not as supplementary

These materials have been moved into the main text (lines 314-331).

Comments on the Quality of English Language:

-Minor editing of English language required

The manuscript has been reviewed to eliminate punctuation inaccuracies and to make the text more fluid, especially in the materials and methods section. Authors will be happy to further address any additional specific grammatical errors.

Round 2

Reviewer 3 Report

Comments and Suggestions for Authors

The authors have made substantial improvements in the revised version, rendering it suitable for publication.

Comments on the Quality of English Language

Minor editing of English language required.